# Sustainable Ecocements: Chemical and Morphological Analysis of Granite Sawdust Waste as Pozzolan Material

**DOI:** 10.3390/ma13214941

**Published:** 2020-11-03

**Authors:** Santiago Yagüe, Cristina González Gaya, Victor Rosales Prieto, Alberto Sánchez Lite

**Affiliations:** 1Department of Construction and Manufacturing Engineering, ETSII, National Distance Education 5 University (UNED), C/Juan del Rosal 12, 28040 Madrid, Spain; cggaya@ind.uned.es (C.G.G.); victor.rosales@ind.uned.es (V.R.P.); 2Department of Materials Science and Metallurgical Engineering, Graphic Expression in Engineering, Cartographic Engineering, Geodesy and Photogrammetry, Mechanical Engineering and Manufacturing Engineering, School of Industrial Engineering, Universidad de Valladolid, Paseo del Cauce 59, 47011 Valladolid, Spain; asanchez@eii.uva.es

**Keywords:** waste, granite saw dust, pozzolanic reaction, pore size, ecocement

## Abstract

The processes focused on stone cutting generate a large volume of waste. Small size waste, silt/clay, is not used and goes to landfill. However, the composition of these wastes makes them useful for adding to cements and for use in construction. In the present paper, 10% Ordinary Portland cement is replaced by 10% waste from granite sawmill, which is studied to obtain sustainable ecological cement. This replacement provides advantages from the morphological and chemical point of view at the cements. The waste has a particle size that does not exceed 15 µm and that when replacing in the cement, after the hydration reaction, generates structures where Calcium Silicate Hydrate (C-S-H) gels and double layered hydroxide compounds (LDH) are reaction products formed in high concentration. These products develop stable phases in the structures over long time periods such one year, which was the time frame used in this study.

## 1. Introduction

Research and eco-innovation for new alternative sources of raw materials from waste are established as possible actions aimed at achieving greater efficiency in the use of resources. Solid waste, if not managed properly, according to the principles of prevention, reuse and recycling, tend to accumulate, with environmental, economic and social impacts. In the countries of the European Union (EU), not all Construction and Demolition Waste (CDW’s) have the same applications. The properties of the recycled waste, from mixed construction and demolition waste (masonry and mortar) are highly variable, and this minimizes the use of said material. The development of classification techniques capable of reducing this variability is essential for quality control purposes and for the production of ecological high-value materials. The use of these wastes is dependent, among other properties, on the waste porosity for its inclusion in the international standards of new cement, which will condition their mechanical properties [1].

The cement industry, in peak production years (55 Mt of cement) used up to 5.7 million tons of different by-products and industrial waste such as clinker additions (blast furnace slag, fly ash, silica fume, industrial pozzolans activated), reducing to less than half in 2019. These industrial pozzolans are mainly aimed at the manufacturing of binary cements (clinker + pozzolan, type II A/B) as a consequence of energy and economic savings, over type I cement without additions.

The use of alternative materials in cement production is limited by the need for their availability in large quantities [2]. The sources of material for the production of cement are not renewable, and it is at this point where the addition of recycled products must be taken into account for its sustainability [3].

However, there is growing concern in the Spanish cement industry about the progressive decrease in the availability pozzolans traditionally listed in the European standard. In this context, the need to diversify the sources of mineral additions with pozzolanic activity for use in cement manufacture and derived materials is highlighted [4,5,6].

One of the most interesting CDW’s for its use are those of the ornamental stone industry, in general, and in particular, the one whose raw material is carbonate rocks, marble type [7,8,9,10,11,12,13]. The use of this waste in areas close to its generation favours its application, by reducing the economic costs of transport. Other types of abundant ornamental rock wastes are the ones with a siliceous composition. These wastes (granodiorite or basalt) improve resistance to freezing/thawing and susceptibility to reaction with alkalis [14]. Ramos et al. [15] have worked on granite mining waste, considering that, if the wastes are ground with sufficient finesse, it produces a denser matrix that promotes a reduction of up to 38% in the expansion of the new mortar obtained by adding this waste to the cement and a resistance to chlorides of almost 70%, without compromising workability and resistance.

The incorporation of granite wastes in the cement decreases the resistance to compression and flexion, but it provides better results in water absorption, abrasion and water permeability. Concrete containing granite waste, up to 20%, could be recommended for all construction applications and a substitution of between 20–40% would be useful, for example, in pavements [16,17,18,19]. It is feasible to use the waste generated by cutting and finishing of granite blocks as a replacement for fine aggregate in mortar mixes (30% to 40%), giving better results than conventional mortars [20]. Mashaly et al. [21] have suggested substitutions of cement for granite waste by up to 40%. The results showed that the modified mixes with granite up to 20% cement replacement showed an insignificant decrease in physical and mechanical properties compared to cement without substitution, in addition to greater resistance to abrasion, freezing/thawing and sulphate attack [22]. Jain et al. [23] have studied the properties of fresh and hardened concrete through various tests with 20% substitution of granite rock waste. Also, red granite powder can be used to replace up to 30% by mass in mortars. The resulting materials were adequate in properties such as workability, stiffness, and strength compared to standard Portland cement [24]. A material is a pozzolan when all the reactions and processes that occur between the active pozzolan components (especially silica and alumina), calcium hydroxide and water are adequate. The products obtained are similar to those of the hydration of Portland cement [25,26,27]. The present paper studies the possibility of 10% substitution of the waste from a granite sawmill in an ordinary Portland cement to obtain a sustainable, ecological cement that takes advantage of the waste from the granite cutting industry that until now has generated large volumes without a specific use.

## 2. Materials and Methods

### 2.1. Materials

The materials have been selected and collected at the “Granitos Cardeñosa” granite sawmill, in Cardeñosa (Ávila, Spain), where slabs of this material are cut with a large volume of rock sawdust (the generation of these wastes is around 12 tons/year, which up to now have had no utility and produced an inert but bulky waste). As the cutting is carried out wet, the sludge obtained is allowed to dry in the open air (P sample) and the material of the bush hammering of grey granite (PR-1 sample) and blond granite (PR-2 sample) is collected dry. A commercial ordinary Portland cement (CPO) type CEM I 42.5 R is used for the substitutions (Table 1). This cement is characterized by having a mass composition between 95–100% Clinker and 0–5% minority components [28]. The sand used in the study has been CEN EN 196-1 standard [29], natural sand with a silica concentration higher than 98%.

### 2.2. Methods

Particle size was determined by laser diffraction spectrometry (DRL) in a range of 0.1 µm to 1750 µm (SYMPATEC HELOS 12 LA spectrometer, Labman, North Yorkshire, UK).

Analysis for the identification of solid phases has been performed by X-Ray Diffraction (XRD) on an X-ray diffractometer model SIEMENS D-5000 (Siemens, Madrid, Spain). The disoriented powder diffractogram has been recorded from 5 to 60 degrees with a sweep rate of 2 degrees per minute. In the X-ray generator tube, a tungsten filament is used and a copper plate (Cu Kα) as the anode. The current and voltage intensity applied to the X-ray generator tube are 30 mA and 40 kV, and the dispersion and reception slits are 1 and 0.18 degrees, respectively. The identification and quantification uses the Match v.3 and Rietveld Full Prof software (Match!) Program with the Inorganic Crystal Structure Database (ICSD) and the Crystallography Open Database (COD) [30], with rutile as internal standard.

The equipment used for Scanning Electron Microscopy (SEM/EDX) analysis is a scanning electron microscope PHILIPS model XL (FEY, Hillsboro, OR, USA) with tungsten source. The samples were fixed to the metallic sample holder by means of an adhesive sheet on both sides of graphite. The surface of which was later metallized with gold to guarantee conductivity, in a BIO-RAD model SC 502 equipment. The same electron microscopy equipment allows for spot chemical analysis, by X-ray dispersive energy with silicon/lithium detector and X-Ray Dispersive Energy (EDX) DX4i analyser (FEY, Hillsboro, OR, USA).

To determine the pozzolanic activity of the materials, an accelerated method is used based on the pozzolanicity test of cements [31], which is based on the reaction of the pozzolanic material with the lime released during the cement hydration. The sample is introduced into the lime solution for 1, 7, 28, 90, 180, and 360 days. Once this period of time has elapsed, at 40 °C, the solution is titrated to obtain the real concentrations of Ca^2+^ and OH^−^ ions, previously filtered as quickly as possible, to avoid carbonation, with a filter of pore size 240. Take 75 mL of this filtrate and 1 g of sample, in hermetically sealed polyethylene bottle to avoid losses. The mixture is vigorously shaken and the flasks are left in an oven at 40 °C, which is the desired time to proceed with the titration of the Ca^2+^ and OH^−^ ions. The one related to OH^−^ ions is carried out with ClH, using methyl orange as an indicator and that of Ca^2+^ with EDTA and as calcein indicator. The chemical characterization of the solutions was performed with an Inductively Coupled Plasma Mass Spectrometer (ICP/MS), Perkin Elmer model Elan 6000 with an AS91 automatic injector (Perkin Elmer, Madrid, Spain).

## 3. Results and Discussion

Chemical analysis of the CEM I 42.5 R Ordinary Portland Cement (CPO) indicated that the main component was CaO, followed by silica and, to a lesser extent, aluminum, sulphur and iron oxides (Table 1). If it is assumed that it is made up exclusively of clinker, its potential composition is C_3_S 67.4%, C_2_S 8.6%, C_3_A 9.28% and a ferritic phase (C_4_AF 9.97%), calculated by the content of its main oxides, using the Bogue equation [32].

The granite sawdust, rich in silicates (quartz, feldspars, phyllosilicates), provides CPO (rich in alkalis), with a composition where Si and Al are the main components, which improves the properties of cements, such as workability, thus generating a pozzolanic cement of an ecological nature. Cement have been introduced into the manufacturing cycle of waste and recycled materials that replace those extracted from natural sources and that, in some cases, may almost be missing. The chemical analysis of granite wastes by ICP/MS is presented in Table 1 for the main components. The minor elements are lithium, boron, copper, rubidium, strontium, and barium in concentrations above 100 ppm and variable amounts of tungsten, lead, and copper, among others. The sum of the oxides of silicon, aluminum and iron is next to 90%, in addition, alkaline elements contents is bigger than 5%. The acid composition along with the particle size indicates significant pozzolanic activity.

The chemical composition of the studied materials is similar to that described by different authors [15,24,33,34,35,36,37], among others, in waste from various sources, but similar in nature.

Mineralogical composition of all samples is similar, with variable contents for albite and orthoclase (Figure 1 and Table 2). PR-2 sample lacks kaolinite and has a low concentration of biotite. PR-1 sample has the highest content of amorphous material (10%).

Natural stone corresponds to the granite of two micas (muscovite and biotite) that have in their composition K-feldspar (microcline, orthoclase), quartz, CaNa-feldspar (anorthite altered to sericite) and rutile, and as accessory minerals apatite, zircon, sphene, chlorite, and tourmaline, and whose thin sheet reproduces the aforementioned minerals (Figure 2).

For the use of a material such as pozzolan, the particle size is very important. The particle size distribution was made by DRL between 0.1 µm and 1750 µm. The particle size distribution curves are similar for all samples, with a maximum value of 9.5 μm for P sample and a 5 μm for PR-1 and PR-2 samples (Figure 3). The results indicate that all the samples can be considered possible pozzolans from the particle size. The fineness of the material generates a disintegration of crystalline structures with a high specific surface that facilitates the pozzolanic reaction [38].

The largest size is 9.5 µm for P sample and, coinciding at 5 µm, for PR-1 and PR-2 samples. These latter samples have greater homogeneity.

### 3.1. Waste/Lime System

In order to know the behavior of the wastes, the waste/lime system has been studied by means of the pozzolanic activity test; the results of the liquid phase at 1, 7, 28, 90, 180, and 360 days of reaction are found in Figure 4.

The samples have pozzolanic activity for all reaction times (Figure 4). Its trend is logarithmic, with an increase at 90 days of reaction. The sample with highest fixed lime is the PR-1. The study of the kinetics reaction of the pozzolanic reaction is followed by the study of the hydrated phases that appear in the solid phase and that will determine the acceptance or rejection of these wastes as additions in the CPO composition.

The identification of solid phases has been carried out by XRD analysis. Using this technique, newly formed compounds have been identified at the different reaction times (1, 7, 28, 180, and 360 days) (Figure 5).

The reaction phases are LDH compounds, hydrated calcium aluminate (C_3_AH_13_), calcite, stratlingite, and low crystallinity CSH gels (Table 3). In addition, the initial minerals appear from the sample (quartz, feldspars and phyllosilicates). This result is discordant with that proposed by Medina et al. [37] which, in similar samples, do not mention stratlingite. LDH (phyllosilicate/carbonate) compounds have been cited in the literature as carboaluminates, but in recent times they are considered double oxides, formed by superimposed sheets of a tetrahedral layer of silicon with oxygen and an octahedral layer of aluminum with groups (OH)^−^ kaolinite type (1:1 phyllosilicate) [37,39]. CSH gels are composed with a loosely organized structure, but arranged in nano-sized sheets and organized in chains of a finite number of members. Given their not very crystalline nature, they will be quantified as amorphous material (Table 3). Portlandite comes from the reactants of the pozzolanic reaction that decompose to CaO, due to the manipulation of materials and their reaction with atmospheric CO_2_, following a carbonation process.

Stratlingite is related to the hydration reaction of aluminum-rich cements. In these cements they can coexist with hydrated calcium silicates or CSH gels [40,41]. Peaks appear at 12.61 Å, 6.29 Å, 4.40 Å, 4.15 Å, 2.87 Å, 2.61 Å, 2.50 Å, 2.48 Å and 2.36 Å in reaction times greater than 180 days. Calcium aluminate hydrated at 7.86 Å, 2.88 Å, and 2.86 Å is identified at all ages, in amounts that increase with reaction time. LDH compounds are recognized at 7.59 Å and 7.41 Å for spacing (006) and 3.78 Å for (003) at all ages, their concentration increasing with the reaction time.

All these materials are present together with portlandite, which decreases in concentration over time, and calcite, which remains almost constant. As time passes, the amorphous material increases (18% of concentration after one year of reaction). The new phases appear and increase in concentration at the expense of initial feldspars and phyllosilicates that decrease in quantity (Table 3).

The addition of wastes, pozzolans, in the manufacture of cements shows that portlandite decreases and hydraulic products appear, such as hydrated calcium silicates. In addition, carboaluminates have been identified among other compounds, which contribute to cement having greater durability in carbonation processes, greater resistance to attack by sulphates, less chloride penetration, less acid attack, and arid-alkali reaction.

The morphology of the different species that appear in the samples has been studied using SEM/EDX (Figure 6 and Figure 7). The materials are formed and deposited on pre-existing substrates, generating aggregates of variable compositions. It is recognized that stratlingite, according to surface chemical analyses (Table 4), is identified as hexagonal sheets that are arranged interspersed with CSH gels and LDH compounds. The latter, also of hexagonal morphology, are recognized by their SiO_2_, Al_2_O_3_ and CaO ratio [30], as well as portlandite and hydrated calcium aluminate.

### 3.2. Waste/Mortar System

After studying the waste reactions in the waste/lime system, the waste/cement system will be followed. When the anhydrous CPO diffractogram is observed, calcite is identified, with its typical reflections at 3.03 Å, 2.48 Å, 2.28 Å, 2.09 Å, 1.92 Å, 1.88 Å, 1.62 Å, and 1.60 Å. In addition, alite, tricalcium silicate (C3S), appears as a major component at 5.90 Å, 3.86 Å, 3.51 Å, 3.34 Å, 3.22 Å, 3.04 Å, 2.98 Å, 2.78 Å, 2.75 Å, 2.61 Å, 2.33 Å, 2.19 Å, 1.99 Å, 1.94 Å, 1.83 Å, 1.76 Å, 1.63 Å, and 1.54 Å. Along with alite, belite, dicalcium silicate (C2S) is also recognized at 2.75 Å, 2.69 Å, 2.19 Å, 2.03 Å and 1.93 Å; bassanite at 5.94 Å, 3.45 Å, 2.98 Å, 2.78 Å, 2.12 Å, and 1.84 Å; ferritic phase (C3AF) at 7.26 Å, 3.64 Å, 2.66 Å, 2.64 Å, 2.05 Å, 1.92 Å, and 1.81 Å; tricalcium aluminate at 2.71 Å, 2.69 Å, 2.29 Å, 1.92 Å, 1.91 Å, and 1.89 Å; in addition to portlandite, ettringite and tetracalcium aluminate (C4AF). The quantification of all phases is provided in Table 5.

Considering the variation that the CPO undergoes in the different curing times, it can be said that, in all cases, calcite appears, in a proportion that remains almost constant, with a slight increase in this mineral over long periods of times. The justification for the persistence of calcite is due to carbonation of the portlandite generated in the handling process and environmental carbonation.

In short curing times, 1 and 7 days, the X-ray diffraction reflections of the bassanite disappear, and with time, in the ferritic phase, tricalcium aluminate and alite intensities decrease, the latter becoming almost imperceptible after a year. All the reactive phases of the anhydrous CPO minimize their concentration with respect to the curing time, which indicates the correct development of the expected reaction.

Belite has an almost constant concentration in the process up to 90 days of reaction, and from that moment it decreases, which leads again to think about its transformation into hydrated products. Portlandite appear first with reflections at 4.90 Å, 3.11 Å, 2.62 Å, 1.92 Å, 1.79 Å, and 1.68 Å, increasing in concentration with time. CSH gels at 3.07 Å, 2.80 Å and 1.83 Å, with low crystallinity, are also recognized. After 1 year of reaction, the reflection at 9.80 Å relative to CSH gels type II is identified, confirmed by EDX, understanding with Taylor [30] that, in these CSH gels, the CaO/SiO_2_ ratio is between 1.50 and 2.00, which is recognizable by its appearance under the electron microscope, similar to fibber bundles (Table 6).

Ettringite, a typical material that appears as a hydration product, is also identified in the reflections at 9.73 Å, 5.61 Å, 3.88 Å, 2.56 Å, and 2.20 Å with its maximum development after 7 days of reaction, and then disappears. Tetracalcium aluminate 13 hydrate is poorly defined in the peaks at 8.17 Å and 2.88 Å, with low crystallinity. Finally, LDH compounds are present at 7.60 Å and 7.2 Å, which increase in quantity until reaching a maximum value at 360 days (33%).

By SEM EDX, some of the morphologies of the compounds present in ordinary Portand cement are observed. Thus, belite, alite and portlandite are recognized at time 0, together with aluminate and bassanite (Figure 8A,B, Table 6).

As the reaction time progresses, after a week, new phases appear, such as ettringite, characterized by the long fibbers that form its crystals (Figure 8C). After 1 year, it is the fibrous ettringite crystals and the CSH gel type II bundles that grow on hexagonal portlandite crystals that serve as their substrate (Figure 8D).

Portland cements during their hydration generate CSH gels that initially need high water content for their development, much of which is lost with gentle drying. This circumstance is related to alite hydration, through the Ca^2+^ (aq), fully hydrated, on anhydrous anionic base plates. Both the solvated Ca^2+^, as well as the CaOH^+^, are incorporated to balance the negative charge of the basal lamina, which leads to a significant structural disorder and thus explains the little crystalline nature of CSH gels formed. One part of the solvation water is irreversibly lost in the first drying, due to the existence of Si–O–Ca–O–Si bridges, which justifies the irreversible contraction. CSH gels are metastable and evolve towards more stable phases, forming portlandite and elimination of water [42,43,44].

### 3.3. Waste/Mortar System with Substitution

Diffractograms with the 10% substitution of P, PR-1 and PR-2 wastes are very similar to each other. The phases that are recognized are alite, belite, portlandite, calcite from portlandita carbonation, ettringite, CSH gels, tetracalcium aluminate, LDH compounds, quartz, phyllosilicates, and feldspars; these last three minerals are from the original sample (Figure 9). Comparing these diffractograms with those of CPO, the difference is centered in the appearance of the initial minerals of the wastes that are not identified in the cement. With respect to the waste/lime systems, the absence of stratlingite in the waste/cement system stands out. This difference can be attributed to the interference of the cementing matrix in the crystallization degree of the stratlingite [43]. When comparing the diffractograms with each other, it can be said that the reaction products are the same as those identified in CPO cement studied previously, with the exception that the LDH compounds appear at shorter times, 7 days, compared to ordinary Portland cement, and that its concentration increases as the curing process proceeds. Amorphous material increases in quantity over time. Alite decreases in concentration until it reaches traces after one year of reaction, and belite has its maximum at 28 days, and then decreases until it reaches the starting value.

SEM/EDX analysis allows the identification of the species recognized by X-ray diffraction analysis with portlandite, fibrous ettringite crystals, similar to CSH gels. At 180 days, they show CSH type II gels and abundant LDH compounds that nucleate on portlandite, forming root aggregates (Figure 10). CSH gels show a fibrillary morphology that is confused with that of LDH compounds, however, they are differentiated by means of specific chemical analysis by EDX, where the CaO/SiO_2_ ratio is close to 2.00 [45,46,47,48]. The existence of ettringite does not seem to be related to the cracking defects generated in mortars [49]. The disappearance of ettringite with the curing time is essential to avoid the appearance of cracks or fissures [50].

## 4. Conclusions

Three samples of granite sawdust from Cardeñosa ornamental stone sawmill (Ávila, Spain) have been studied. They are siliceous rocks whose composition is formed by quartz, biotite, kaolinite, amorphous material, and feldspars (albite and orthoclase). The size of the particles is between 5 µm and 9.5 µm.

By chemical analysis all the samples are rich in silica, alumina, iron oxides, and alkaline elements, in short, an acid composition.

Pozzolaniticy tests have been carried out with natural samples. After 1 year of reaction and monitoring at times 1, 7, 28, 90, 180, and 360 days, all samples have pozzolanic activity, with the highest values for gray granite sawdust, which contains a lot of amorphous material. In the characterization of the solid, different hydrated phases are present, which are generated from the initial materials and remain almost unchanged, and those from the reaction with calcium hydroxide. LDH compounds, hydrated calcium aluminate, calcite, stratlingite, and low crystallinity CSH gels have been identified. Although the new phases appear at all times from the initial moment to the year of reaction, they do so in different proportions, with a special mention of the increase in amorphous materials, as time passes. Amorphous materials and CSH gels, by SEM, are visualized as aggregates that act as crystallization nuclei of the new species.

The difference with residue/lime system is the absence of stratlingite and the appearance of ettringite, with the same reaction products in the CPO, except that the LDH compounds are formed in shorter times than in the case of the initial CPO without substitution and that the amount of CSH gels increases with time; in addition, the hydration reaction occurs. The gels are type II in the Taylor classification, which are identical to those obtained in the case of CPO without substitution.

When 10% waste is added to the CPO, the same compounds are produced as if it were cement without addition, and they coexist with the initial minerals in the waste (quartz, feldspar, mica, kaolinite, and hematite). CSH gels and LDH compounds formation appears in great quantity, but portlandita behaviour is different. The portlandite decreases in t medium times and remains in long ones. Granite sawdust wastes are suitable to be added to Ordinary Portland Cement, thus avoiding the accumulation of these wastes for the preparation of pozzolanic cement and promoting its sustainability.

Studies on mechanical properties as well as their application to specific problems such as the development of slab track are being carried out in the long term, with satisfactory results, up to now, according to the applicable standards.

## Figures and Tables

**Figure 1 materials-13-04941-f001:**
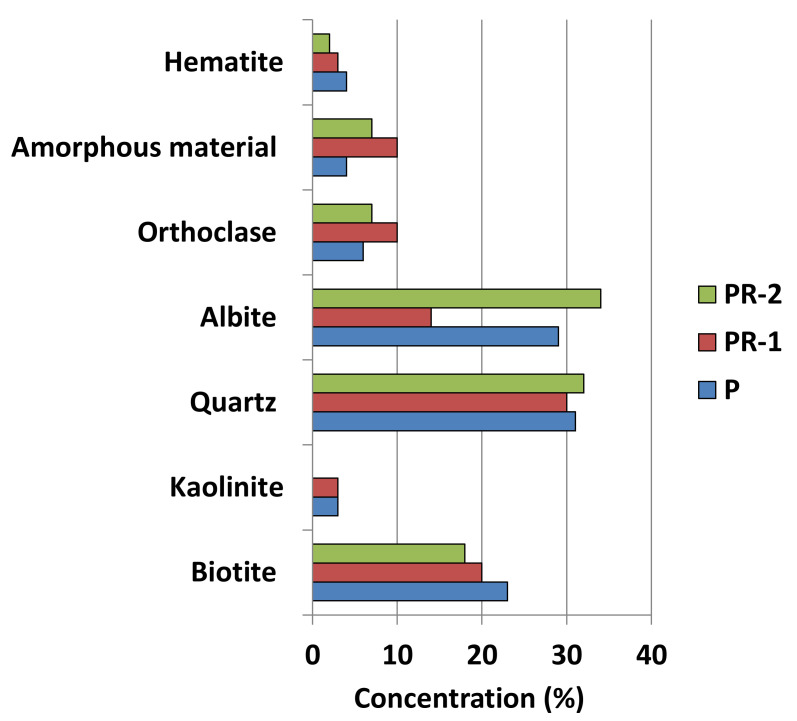
Minerals concentration by XRD diffraction from natural samples.

**Figure 2 materials-13-04941-f002:**
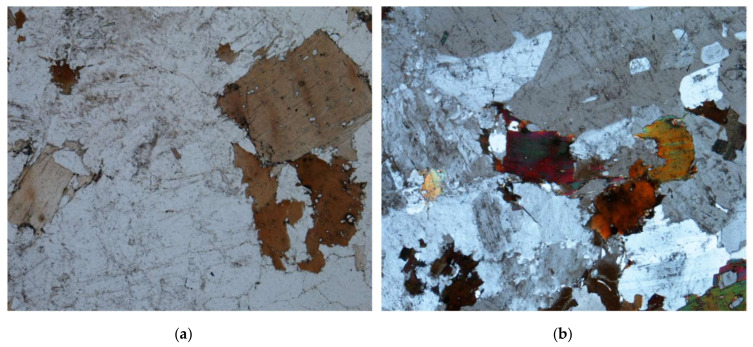
(**a**) Big biotite crystal with zircon inclusions and feldspar crystals (white light X64). (**b**) Quartz altered feldspars and biotite crystals (polarized light X64).

**Figure 3 materials-13-04941-f003:**
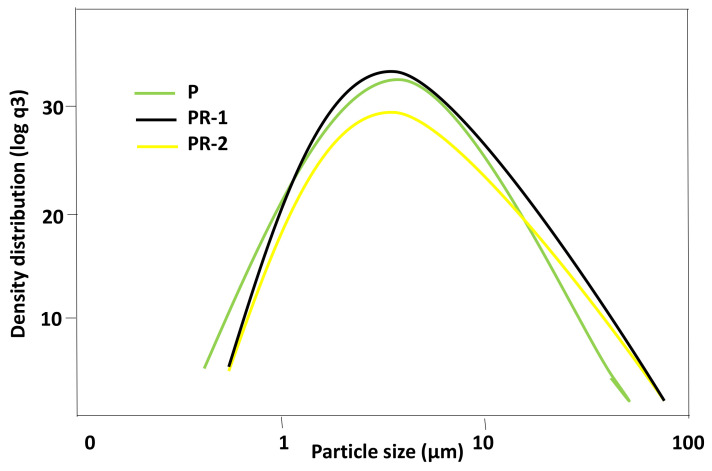
Distribution by particle size from the studied samples.

**Figure 4 materials-13-04941-f004:**
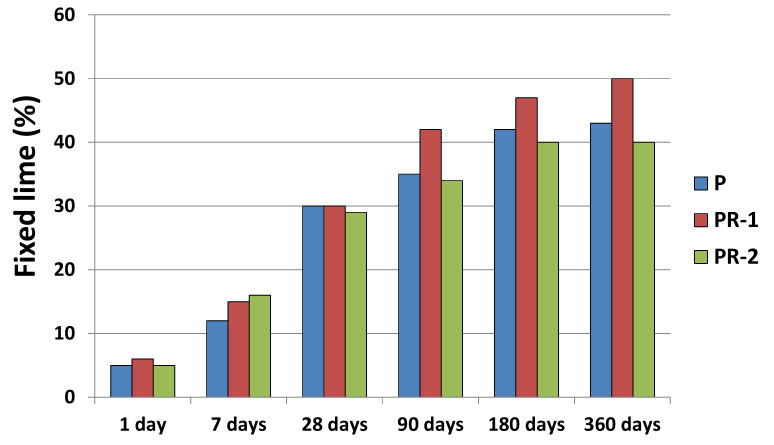
Fixed lime analysis for the samples up to 1 year of reaction.

**Figure 5 materials-13-04941-f005:**
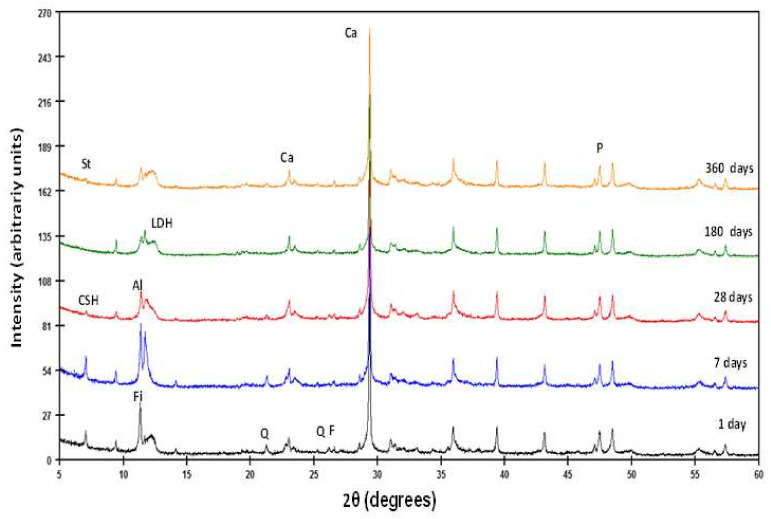
XRD patterns of solid phases from waste/lime system. Q—quartz; F—feldspars; Fi—phyllosilicates; Layered Double Hydroxides (LDH) compounds; Al—aluminate; St—stratlingite; Ca—calcite; P—portlandite.

**Figure 6 materials-13-04941-f006:**
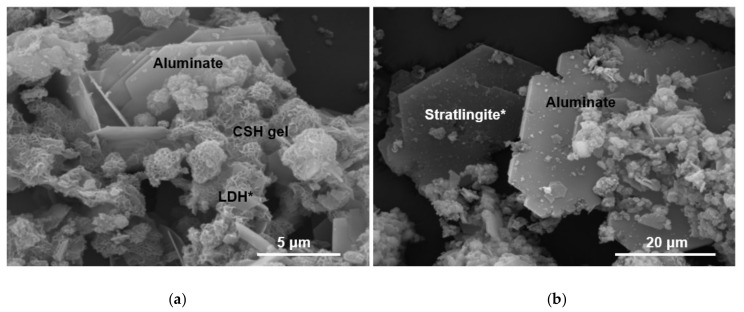
Solid phases present at 90 days of hydration reaction (*indicates the analysis point listed in Table 4). (**a**) Aluminate, C-S-H gels and LDH compounds; (**b**) Stratlingite and aluminat.

**Figure 7 materials-13-04941-f007:**
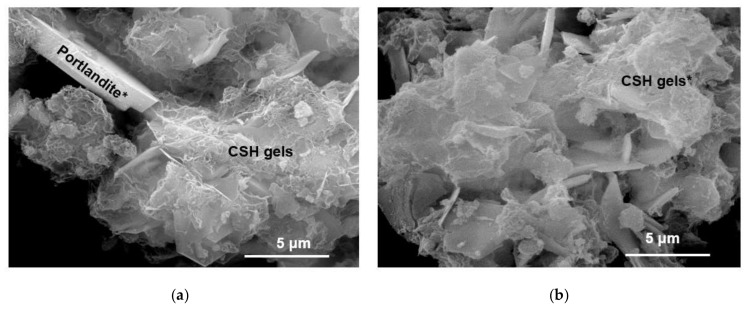
Solid phases present at 360 days of hydration reaction (*indicates the analysis point listed in Table 4). (**a**) Portlandite and C-S-H gels; (**b**) C-S-H gels.

**Figure 8 materials-13-04941-f008:**
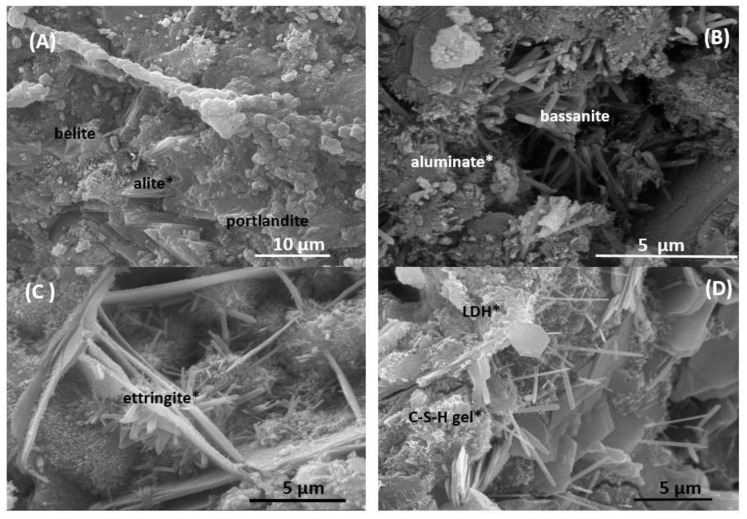
Reaction phases from CPO. (**A**,**B**) CPO at 0 days. (**C**) Ettringite at 7 days. (**D**) C-S-H gel and LDH at 360 days (* indicates the analysis point listed in Table 6).

**Figure 9 materials-13-04941-f009:**
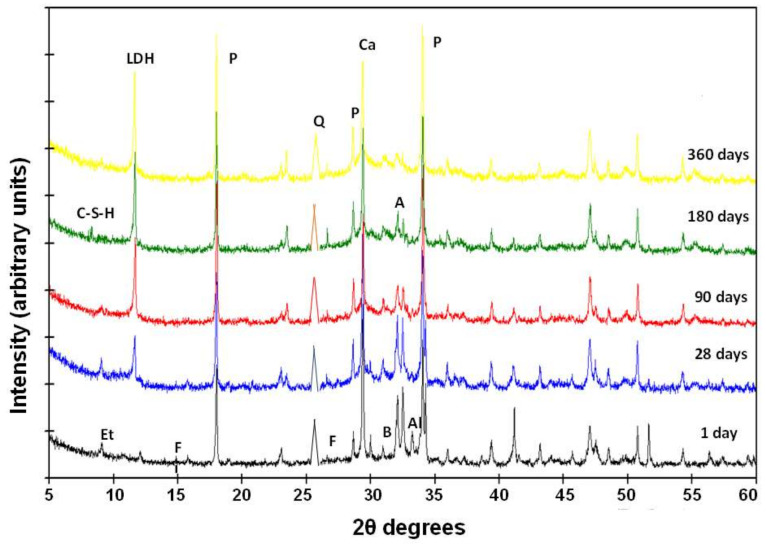
XRD patterns of the CPO with 10% addition of waste cured at 1, 28, 90, 180, and 360 days. P—portlandite; E—ettringite; LDH—LDH compound; Ca—calcite; A—alite; B—belite; C-S-H—gels; Al—aluminate; Q—quartz; F—feldspar; Fi—phyllosilicate.

**Figure 10 materials-13-04941-f010:**
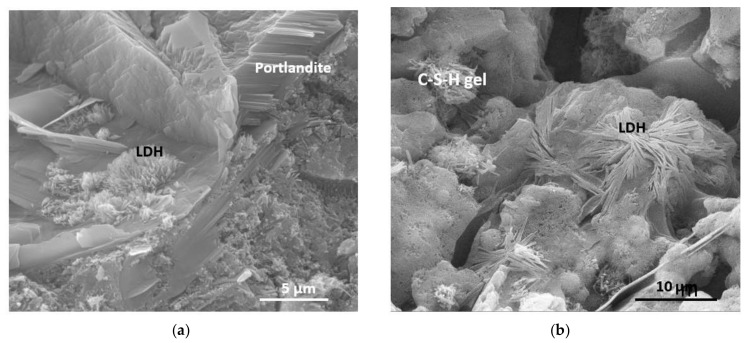
Materials observed by SEM: CSH gels type II and LDH compounds that grow over portlandite hexagonal crystals. (**a**) Portlandite LDH and compounds; (**b**) CSH gel and LDH compounds.

**Table 1 materials-13-04941-t001:** Chemical analysis for the materials in these investigation (n.d.—not detected; LOI—loss on ignition).

Oxides (%)	CPO	P	PR-1	PR-2
SiO_2_	19.20	68.56	78.89	82.08
Al_2_O_3_	5.62	18.81	13.03	13.64
Fe_2_O_3_	3.08	1.95	2.09	0.86
CaO	62.44	1.65	1.47	0.52
MgO	1.21	0.39	0.19	0.06
SO_3_	3.29	n.d.	n.d.	n.d.
K_2_O	0.89	4.14	1.10	1.05
Na_2_O	0.27	4.30	1.36	1.73
MnO	n.d.	0.03	0.03	0.03
TiO_2_	0.24	0.20	0.25	0.06
P_2_O_5_	0.11	n.d.	n.d.	n.d.
LOI	2.72	1.12	0.47	0.42

**Table 2 materials-13-04941-t002:** Quantification Rietveld method (R_B_ and Χ^2^, agreements factors: n.d.—not detected; B—biotite; C—kaolinite; Q—quartz; A—albite; O—orthoclase; H—hematite).

Sample	B (%)	C (%)	Q (%)	A (%)	O (%)	Amorphous Material (%)	H (%)	R_B_	Χ^2^
P	23	3	31	29	6	4	4	19.4	3.7
PR-1	20	3	30	14	10	10	3	17.3	3.5
PR-2	18	n.d.	32	34	7	7	2	14.3	4.9

**Table 3 materials-13-04941-t003:** Quantification Rieltveld of the solids phases from waste/lime system (R_B_ and Χ^2^, agreements factors; n.d.—not detected; Q—quartz, F—feldspar, Fi—phyllosilicates, P—portlandite, Ca—calcite, S—stratlingite, MA—amorphous material).

Reaction Time (days)	Q (%)	F (%)	Fi (%)	LDH (%)	S (%)	C_3_AH_13_ (%)	P (%)	Ca (%)	MA (%)	X^2^	R_B_
1	25	18	15	4	n.d.	traces	27	6	4	11.2	4.3
7	26	15	12	4	n.d.	traces	24	8	10	13.6	5.8
28	30	14	10	4	n.d.	4	16	10	12	12.6	4.2
90	29	12	8	4	traces	6	13	11	15	13.4	4.2
180	30	9	4	7	traces	10	12	10	16	15.8	3.9
360	30	8	4	10	traces	11	9	6	18	14.3	4.8

**Table 4 materials-13-04941-t004:** Chemical analysis by EDX (* indicates the analysis point from Figure 6 and Figure 7).

Oxides (%)	CSH Gel *	LDH *	Stratlingite *	Portlandite *	Aluminate *
Al_2_O_3_	18.20 ± 3.28	26.04 ± 0.92	18.23 ± 0.89	-	16.86 ± 0.39
SiO_2_	31.07 ± 0.06	40.39 ± 0.67	16.76 ± 0.79	-	4.12 ± 0.18
CaO	50.32 ± 0.37	31.15 ± 0.89	64.50 ± 2.41	100	76.79 ± 1.34
CaO/Al_2_O_3_	2.79	1.20	3.54	-	4.04
CaO/SiO_2_	1.63	0.77	3.85	-	18.63
SiO_2_/Al_2_O_3_	1.61	1.55	0.92	-	0.21

**Table 5 materials-13-04941-t005:** Quantification Rietveld by anhydrous CPO and at 1, 7, 28, 90, 180, and 360 days (R_B_ and Χ^2^, agreements factors; n.d.—not detected; T—traces. MA—amorphous material; Ca—calcite, A—alite, B—belite, Ba—bassanite, FF—ferritic phase; E—ettringite; P—portlandite; A3—tricalcic aluminate; A4—tetracalcic aluminate).

Reaction Time (Day)	MA (%)	Ca (%)	A (%)	B (%)	Ba (%)	FF (%)	A3 (%)	E (%)	CSH (%)	A4 (%)	P (%)	LDH (%)	X^2^	R_B_
0	5	T	38	28	T	8	n.d.	n.d.	n.d.	18	n.d.	n.d.	17.2	4.6
1	17	T	14	30	T	8	T	T	T	T	23	n.d.	15.2	5.2
7	13	T	8	31	T	4	T	6	T	T	34	T	16.3	4.8
28	7	4	4	28	n.d.	T	T	T	T	T	37	13	17.5	3.9
90	9	4	T	25	n.d.	3	T	n.d.	T	T	38	18	13.1	5.2
180	10	4	T	18	n.d.	2	T	n.d.	T	T	36	24	13.2	5.9
360	9	4	T	14	n.d.	2	T	n.d.	T	T	46	33	14.8	3.9

**Table 6 materials-13-04941-t006:** Chemical analysis by EDX (* indicates the analysis point from Figure 8; n.d.—not detected).

Oxides (%)	C-S-H Gel *	LDH *	Alite *	Aluminate *	Ettringite *
Al_2_O_3_	8.70 ± 0.40	13.09 ± 0.80	4.30 ± 0.40	21.00 ± 1.00	39.00 ± 1.00
SiO_2_	20.90 ± 0.80	26.92 ± 0.61	30.00 ± 1.00	2.03 ± 0.09	24.00 ± 2.00
CaO	59.00 ± 2.03	54.00 ± 1.04	61.21 ± 0.92	60.12 ± 2.06	33.17 ± 2.08
MgO	1.60 ± 0.40	n.d.	1.60 ± 0.10	n.d.	n.d.
SO_3_	n.d.	n.d.	n.d.	3.91 ± 0.45	2.22 ± 0.32
Fe_2_O_3_	9.52 ± 0.21	2.11 ± 0.45	2.65 ± 0.62	0.92 ± 0.11	1.61 ± 0.25

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
