# Peer review of "Sustainable Ecocements: Chemical and Morphological Analysis of Granite Sawdust Waste as Pozzolan Material"

_materials, 2020, doi:10.3390/ma13214941_

Round 1

Reviewer 1 Report

The research in the article should be enriched with strength tests of cement samples made of the tested mixtures. The authors presented the chemical composition and structure of cement for each of the tested samples. Strength results carried out in accordance with the standards used in construction would be a valuable supplement to their research. The use of granite waste (sawmill) may affect the strength properties of cement. The authors showed a change in the cross-linking of cement samples made with the use of granite waste. The combination of both of these facts would significantly enrich the utilitarian value of the research described.

Suggestion for the Authors that the next article devoted to cement additives should contain the results of strength tests carried out at the same time intervals. Comparison of the test results of a sample containing no described additives with samples spiked with waste. Tests should be carried out in accordance with the guidelines contained in construction standards.

Author Response

Reviewer 1:

We thank the reviewer for all suggestions that have enriched the work. All modifications and responses are presented in red.

The research in the article should be enriched with strength tests of cement samples made of the tested mixtures. The authors presented the chemical composition and structure of cement for each of the tested samples. Strength results carried out in accordance with the standards used in construction would be a valuable supplement to their research. The use of granite waste (sawmill) may affect the strength properties of cement. The authors showed a change in the cross-linking of cement samples made with the use of granite waste. The combination of both of these facts would significantly enrich the utilitarian value of the research described.

Suggestion for the Authors that the next article devoted to cement additives should contain the results of strength tests carried out at the same time intervals. Comparison of the test results of a sample containing no described additives with samples spiked with waste. Tests should be carried out in accordance with the guidelines contained in construction standards.

Response to reviewer 1:

Thank you very much for your suggestion. The mechanical tests have been carried out on the mixtures of both 10% and 20%, however in this first work only the Chemical and morphological characterization is contemplated, as mentioned in the title.

Incorporating all the mechanical tests would give the paper a disproportionate length for this publication. I can anticipate that the flexural and compression resistance tests have been very satisfactory and are within the norm.

We accept your suggestion that it will enrich the next article by incorporating the strengths of the mixes for one year and comparing them with a standard cement without additives (CEM I 42.5 R).

Reviewer 2 Report

  1. The title is not concise, some phrases “such as” should not be presented in the title;
  2. Abstract is quite allusive and does not provide in depth information of the objectives achieved in this study;
  3. More content on the future work of this study should be added to the conclusion section
  4. Some English errors are identified; please recheck;
  5. Why do not the authors do mechanical tests to characterize the mechanical behaviour of samples?

Author Response

Reviewer 2

We thank the reviewer for all suggestions that have enriched the work. All modifications and responses are presented in red.

The title is not concise, some phrases “such as” should not be presented in the title;

The title has been changed

Abstract is quite allusive and does not provide in depth information of the objectives achieved in this study;

The abstract has been modified adding the objectives achieved

More content on the future work of this study should be added to the conclusion section

A paragraph has been added in the conclusions on the future work applicable on mechanical properties in the new cements

Some English errors are identified; please recheck;

English has been revised

Why do not the authors do mechanical tests to characterize the mechanical behaviour of samples?

The mechanical tests have been carried out on the mixtures of both 10% and 20%, however in this first work only the Chemical and morphological characterization is contemplated, as mentioned in the title.

Incorporating all the mechanical tests would give the paper a disproportionate length for this publication. I can anticipate that the flexural and compression resistance tests have been very satisfactory and are within the norm.

Reviewer 3 Report

This study attempted the use of granite sawdust in cement. The test design is way more sufficicent with tests lasting a year. The topic is interesting. However, an editing of language is suggested before the publication.

  1. Title, "Analysis of Granite Sawdust Waste Such as Pozzolan Material" Should we use "such" here?
  2. line 18, "In the present paper it 18 is studied the substitution of 10% from Ordinary Portland Cement by the waste from a granite 19 sawmill to obtain a sustainable, ecological cement that takes advantage of residues from the granite 20 stone industry that until now generate large volumes of waste without a specific application." It is too long.
  3. Line 35, "CDW" should give the full description for the first time.

Author Response

Reviewer 3

We thank the reviewer for all suggestions that have enriched the work. All modifications and responses are presented in red.

This study attempted the use of granite sawdust in cement. The test design is way more sufficient with tests lasting a year. The topic is interesting. However, an editing of language is suggested before the publication.

English has been revised

Title, "Analysis of Granite Sawdust Waste Such as Pozzolan Material" Should we use "such" here?

The title has been changed

line 18, "In the present paper it 18 is studied the substitution of 10% from Ordinary Portland Cement by the waste from a granite 19 sawmill to obtain a sustainable, ecological cement that takes advantage of residues from the granite 20 stone industry that until now generate large volumes of waste without a specific application." It is too long.

Abstract is new

Line 35, "CDW" should give the full description for the first time.

It is added